# Quantum corrections to the magnetoconductivity of surface states in three-dimensional topological insulators

Gang Shi[1,2], Fan Gao[1,2], Zhilin Li[1,3], Rencong Zhang[1,2], Igor Gornyi [4,5,6], Dmitri Gutman [7] ✉ & Yongqing Li [1,2,3] ✉

The interplay between quantum interference, electron-electron interaction (EEI), and disorder is one of the central themes of condensed matter physics. Such interplay can cause high-order magnetoconductance (MC) corrections in semiconductors with weak spin-orbit coupling (SOC). However, it remains unexplored how the magnetotransport properties are modified by the high-order quantum corrections in the electron systems of symplectic symmetry class, which include topological insulators (TIs), Weyl semimetals, graphene with negligible intervalley scattering, and semiconductors with strong SOC. Here, we extend the theory of quantum conductance corrections to two-dimensional (2D) electron systems with the symplectic symmetry, and study experimentally such physics with dual-gated TI devices in which the transport is dominated by highly tunable surface states. We find that the MC can be enhanced significantly by the second-order interference and the EEI effects, in contrast to the suppression of MC for the systems with orthogonal symmetry. Our work reveals that detailed MC analysis can provide deep insights into the complex electronic processes in TIs, such as the screening and dephasing effects of localized charge puddles, as well as the related particle-hole asymmetry.

It was established long ago that quantum interference between electrons traveling along pairs of time-reversed self-crossing paths leads to observable effects in conductance, because of enhanced or suppressed backscattering[1–4]. The corresponding weak localization and weak antilocalization (WAL) effects can be detected by measuring the magnetoconductivity (MC) in perpendicular magnetic fields[5,6]. For a weakly disordered 2D system, the MC can be described satisfactorily by the seminal Hikami-Larkin-Nagaoka (HLN) formula[6], in which only the first-order interference effect is taken into account, and possible influences of EEI are not considered.

For 2D electron systems of the orthogonal symmetry class (i.e., with spin-rotational symmetry preserved), Minkov et al.[7] showed that the positive MC can be significantly suppressed by the second-order electron interference and the related EEI effect in the particle-hole channel, when the disorder strength is sufficiently strong. They also found that in a conventional semiconductor-based 2D system, the interaction effect in the particle-particle (Cooper) channel also modifies the MC, but to a much less extent than the former effects[7]. For electron systems of *symplectic* symmetry[8–11], however, no work has been reported so far on the investigation of weak-field MC corrections caused by the second-order quantum interference or the EEI effects.

[1]Beijing National Laboratory for Condensed Matter Physics, Institute of Physics, Chinese Academy of Sciences, Beijing 100190, China. [2]School of Physical Sciences, University of Chinese Academy of Sciences, Beijing 100049, China. [3]Songshan Lake Materials Laboratory, Dongguan, Guangdong 523808, China. [4]Institute for Quantum Materials and Technologies, Karlsruhe Institute of Technology, 76021 Karlsruhe, Germany. [5]Institut für Theorie der Kondensierten Materie, Karlsruhe Institute of Technology, 76128 Karlsruhe, Germany. [6]Ioffe Institute, 194021 St. Petersburg, Russia. [7]Department of Physics, Bar-Ilan University, Ramat Gan 52900, Israel. ✉e-mail: Dmitri.Gutman@biu.ac.il; yqli@iphy.ac.cn

In recent years, three-dimensional (3D) topological insulators (TIs) have emerged as an important type of materials with the symplectic symmetry[8,9]. A tremendous amount of work has been carried out to use the MC in low magnetic fields to probe the surface states and their interplay with the bulk states[12-14]. The experimental data has often been fitted to the HLN equation, and the extracted parameters have provided valuable insights into the nature of electron transport in TIs[15-20]. Nevertheless, it is still unclear whether the HLN equation remains valid for the case of the Fermi level brought close to the Dirac point, which is indispensable for pursuing a number of exotic quantum phenomena in TIs[21-23], since under such a circumstance the transport is no longer in the well-defined weak disorder regime and the EEI effects may play a significant role.

In this article, we show that the standard HLN analysis offers incomplete understanding of the MC in the TI surface states at sufficiently low conductances. Based on the scaling theory, we found that both the second-order interference and the EEI effects in the particle-particle and particle-hole channels can influence the MC. In the dual-gated TI samples we observed that the magnitude of low-field MC can be enhanced by up to about 40%, relative to that in the first-order WAL effect. The prefactor $\alpha$ extracted from the HLN fit decreases with increasing carrier density for both electrons and holes, but at very different rates. The underlying mechanism of the MC enhancement and its gate-voltage dependence is discussed in detail. Implications of the particle-hole asymmetry and enhanced electron dephasing are also addressed.

## WAL effect in TIs

For a 2D system with low disorder (with the dimensionless conductance $g \sim g^{(0)} = k_F l_{tr} \gg 1$, where $k_F$ is the Fermi wavevector and $l_{tr}$ the mean free path), the MC in sufficiently weak magnetic field, $B \ll B_{tr} \equiv \hbar/\left(2el_{tr}^2\right)$, follows the simplified HLN equation[6]:

where $\alpha$ is a prefactor depending on the symmetry and number of transport channels, $\psi(x)$ is the digamma function, $B_\varphi = \hbar/\left(4el_\varphi^2\right)$ is the dephasing field with $l_\varphi = \sqrt{D\tau_\varphi}$ being the dephasing length, $D$ the diffusion coefficient, $\tau_\varphi$ the dephasing time, and $G_0 = e^2/(2\pi^2\hbar)$. The prefactor $\alpha$ takes a value of $-1$ and $1/2$ for the orthogonal symmetry and symplectic symmetry, respectively. For the convenience of this work, the sign of $\alpha$ is reversed in comparison with ref. 6 so that the $\alpha$ value is positive for the WAL. For a 3D TI slab with identical top and bottom surfaces and insulating bulk, one expects $\alpha = 1/2 + 1/2 = 1$.

## Results

### Second-order quantum interference effects

For moderate disorder (i.e., $g > 1$, but not much larger than 1), interference effect of the second loops becomes important. According to our calculations (see Supplementary Note 1) based on the scaling theory, the MC can still be described by Eq. (1), but the prefactor will change to $\alpha = 1 + 1/(\pi g)$ for the 3D TI, leading to the enhancement of MC. This contrasts with the case of weak localization in the systems of orthogonal symmetry class, in which the magnitude of $\alpha$ is reduced by $1/(\pi g)$ by the second-order interference effect, as shown in ref. 7.

### EEI effects

Interaction effects in the particle-particle and particle-hole channels can also introduce MC corrections of the order $1/g$. One is Maki-Thompson (MT) correction in the particle-particle (Cooper) channel, which is given by $\frac{\Delta\sigma_{MT}^{(B)}}{G_0} = -\left(\pi^2/3\right)\gamma_c^2 Y\left(B/B_\varphi\right)$ for $B \ll 2\pi B_T$, where $B_T = k_B T/De$ and function $Y(x)$ is defined in Eq. (1). Here, $\gamma_c$ is the effective interaction constant in the Cooper channel. For long-range Coulomb interaction, this constant approaches, upon renormalization[24,25], the "infrared fixed value" $\gamma_c = 1/\sqrt{\pi g}$. In the weak-field limit, the MT correction takes a form

$$\frac{\Delta\sigma(B)}{G_0} \simeq -\alpha\left[\psi\left(\frac{1}{2} + \frac{B_\varphi}{B}\right) + \ln\left(\frac{B}{B_\varphi}\right)\right] = -\alpha Y\left(\frac{B}{B_\varphi}\right) \qquad (1)$$

$$\frac{\Delta\sigma_{MT}(B)}{G_0} \simeq -\frac{\pi}{3g}Y\left(\frac{B}{B_\varphi}\right) \qquad (2)$$

**Fig. 1 | Basic characterization of a dual-gated BSTS microflake. a, b** Optical image and the cross-section diagram of the device. The scale bar is 15 μm. **c, d** Sheet resistance per square ($R_{XX}$) at $B = 0$ and Hall coefficient ($R_H$) plotted as a function of the top- and bottom-gate voltages. **e** $R_{XX}$ at $B = 0$ and $R_{xy}$ at $B = 1$ T along the symmetric gate-tuning line (dashed lines in panels **c** and **d**). Only the bottom-gate voltage is shown. **f** The upper and lower schematically depict the Fermi level of the top and bottom surfaces at the large positive and negative gate voltages, respectively.

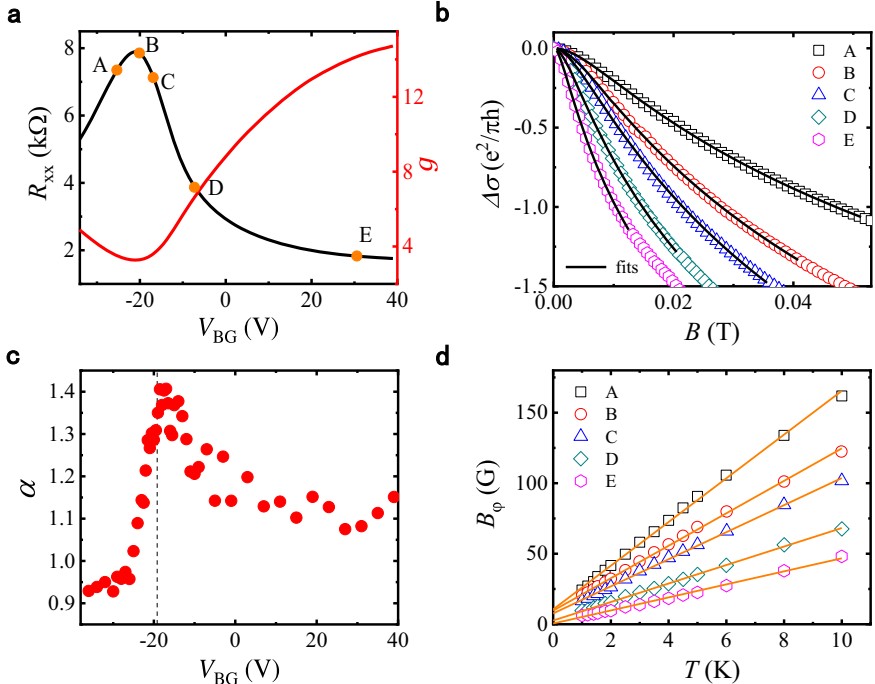

**Fig. 2 | Negative magnetoconductivity (MC) and the HLN analysis. a** Gate-voltage dependence of zero-field $R_{xx}$ and the corresponding conductance $g$ along the symmetric gate-tuning line (see Fig. 1). Points A–E are five sets of gate-voltages selected for detailed analysis. **b** MC curves (symbols) and the fits to the HLN equation (lines). A self-consistent field range of $B = 0-15B_\varphi$ was used. All data shown in panels **a** and **b** were taken at $T = 1.6$ K. **c** Gate-voltage dependence of prefactor $\alpha$ extracted from the HLN fits. **d** $T$-dependences of $B_\varphi$ extracted from the HLN analysis (symbols) and their linear fits for the five gate-voltage points shown in panel (**a**).

At higher magnetic fields, the magnitudes of MT correction will be smaller than the values given in Eq. (2), but the difference is very small when $B$ is comparable with $2\pi B_T$ (Supplementary Fig. S6). On the other hand, the density of states (DOS) correction in the particle-particle channel can also enhance the negative MC substantially when the magnetic length $l_B$ is much shorter than the thermal length $l_T$, approximately corresponding to $B \gg 2\pi B_T$. In comparison, the MT correction is more dominating in lower magnetic fields, which cover a significant portion of the typical fitting range ($B = 0-15B_\varphi$). In the particle-hole channel, the interplay[26] between the long-range (Coulomb) EEI and WAL effects leads to a logarithmic "cross-term" MC correction $\Delta\sigma_{CWAL}(B)$ of the order $1/g$, but with a sign opposite to that of the Cooper channel EEI effects and the second-loop interference corrections. Importantly, this correction, which is expected to follow a $\ln B$ dependence at $B \gg B_\varphi$, saturates at the magnetic fields below $B_d \sim \sqrt{\frac{h}{ed^2}}$ (corresponding to a length scale $l_B \sim d$) due to the presence of the screening by metallic top-gate, where $d$ is the distance to the gate. Details of the calculation of interaction-induced corrections and discussion of the relevant length scales can be found in Supplementary Notes 1 and 5.

Generally speaking, the MC of TI surface states is a rather complicated function of $B$, because the contributions of the aforementioned second-loop interference and EEI effects are associated with multiple length scales, including $l_B$, $l_\varphi$, $l_T$, and $d$, even when $B \ll B_{tr}$. The HLN equation is therefore expected to fail to some degree when the disorder is sufficiently strong and the magnetic field range used for the fitting is wide enough. In particular, for $l_B < l_\varphi$, a natural scale of magnetic field appears, $B \sim 2\pi B_T$, at which the MT correction gets replaced by the DOS correction in the Cooper channel[7] (see also Supplementary Notes 1 and 5). As a result, the MC cannot be described strictly by the HLN expression with a single prefactor $\alpha$. In other words, using the conventional HLN fit would yield a dependence of $\alpha$ on the range of magnetic field where the fit is performed.

## Characterization of dual-gated TI devices

Our transport measurements were carried out on dual-gated $(Bi,Sb)_2(Te,Se)_3$ (BSTS) microflakes. A typical device (Sample A) is depicted in Fig. 1a, b. The bottom surface of the BSTS flake is in contact with a 300-nm thick $SiO_2$ layer, which serves as the bottom-gate dielectric. The top surface is covered with an exfoliated hexagonal-BN layer, which functions as the top-gate dielectric and protects the sample from degradation (more details in "Methods"). Unless otherwise specified, the data presented below was collected from this device. Figures 1c and d show the longitudinal resistance per square ($R_{XX}$) and the Hall coefficient ($R_H$) as a function of the top-gate voltage ($V_{TG}$) and the bottom-gate voltage ($V_{BG}$) at $T = 1.6$ K. The $R_{XX}$ maximum, corresponding to the Fermi level near the charge neutral point (CNP), is located near the center of a roughly diamond-shaped region in Fig. 1c. When $V_{TG}$ and $V_{BG}$ are varied simultaneously along the symmetric gating line (i.e., dashed line passing the two $R_H$ extrema and the $R_{XX}$ maximum in Fig. 1d), the carrier densities on the top and bottom surfaces are expected to be equal.

All the data presented below were taken with the gate-voltage sets along this symmetric tuning line. Plotted in Fig. 1e are the gate-voltage dependences of $R_{XX}$ at $B = 0$ and $R_{XY}$ at $B = 1$ T. $R_{XX}$ reaches the maximum value at $V_{BG} \approx -20$ V, which is not far away from the $R_{XY}$ minimum and maximum (located at $V_{BG} = -24$ V and $-13$ V, respectively). In the region between the two $R_{XY}$ extrema, the chemical potential is close to the CNP, and electrons and hole puddles coexist in the surface states due to electrostatic potential fluctuations[27]. Outside this ambipolar region, the transport is dominated by either electrons or holes. It is noteworthy that the carrier density dependence of $R_{XX}$ suggests that the long-range Coulomb interactions plays an important role in the transport process (see Supplementary Fig. S5).

## Enhanced MC and the HLN analysis

Figure 2a shows the gate-voltage dependence of the dimensionless conductance, which is evaluated as $g = \sigma/(e^2/h)$, with $\sigma$ being

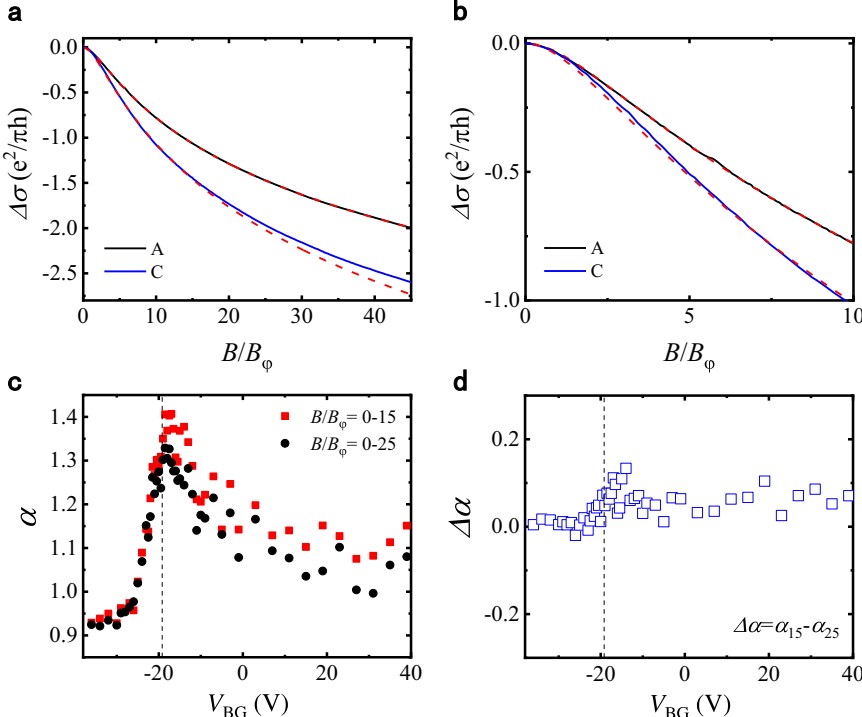

**Fig. 3 | Deviation of the MC from the HLN equation. a** MC curves (solid lines) and the corresponding HLN fits (dashed lines) with fitting range of $B/B_\varphi = 0 - 15$. The fitted curves were extended to $B/B_\varphi = 45$ using the same parameters. **b** HLN fits (dashed lines) of the MC curves (solid lines) for the fitting range of $B/B_\varphi = 0 - 25$. Only the data for $B/B_\varphi = 0 - 10$ is shown for better clarity in fitting quality. **c** Values of $\alpha$ obtained with fitting ranges of $B/B_\varphi = 0 - 15$ (squares) and 0–25 (circles). **d** Difference in the $\alpha$ value between the two fitting ranges depicted in panel **c**, plotted as a function of the gate voltage.

the total conductivity of the top and bottom surfaces at $B = 0$. Depicted in Fig. 2b are MC curves for five sets of gate voltages (labeled as points A–E in Fig. 2a) measured at $T = 1.6$ K, with the MC defined as $\Delta\sigma(B) = \sigma_{XX}(B) - \sigma_{XX}(0)$. All MC curves can be fitted with Eq. (1) by using $\alpha$ and $B_\varphi$ as free parameters. The extracted value of $\alpha$ reaches a maximum value of about 1.4 at $V_{BG} = -18.5$ V, and drops towards $\alpha = 1$ when the Fermi level is tuned away from the CNP (Fig. 2c). The obtained dephasing field $B_\varphi$ exhibits a linear temperature dependence for all cases (Fig. 2d). This is consistent with the theory of EEI-induced electron dephasing[28], which predicts that the dephasing rate $1/\tau_\varphi$ is proportional to the temperature. Using the formula $(k_B T \tau_\varphi/\hbar)\ln(k_B T \tau_\varphi/\hbar) = g$ and the slopes of the $B_\varphi$ vs. $T$ curves in Fig. 2d, the Fermi velocities at points D and E are evaluated to be $4.2 \times 10^5$ and $4.0 \times 10^5$ m/s, respectively (see Supplementary Note 8), consistent with previous angular resolved photoemission (ARPES) measurements on BSTS[29,30].

**Deviation of the MC from the HLN equation**
As illustrated in Fig. 3a, b, the HLN formula provides a good description of the MC data of point C (on the electron side) in the low-field region, but deviates slightly from the experimental data when the magnetic field range is expanded. In contrast, the MC curve for point A (on the hole side) can be satisfactorily fitted to the HLN equation for a wide range of magnetic fields. Nevertheless, different fitting ranges produce the overall similar gate-voltage dependences of $\alpha$ despite some quantitative differences, as depicted in Fig. 3c. For both fitting ranges ($B/B_\varphi = 0-15$ and 0–25), $\alpha$ reaches maximum near the CNP and exhibits an asymmetry between the electron and hole sides. Interestingly, a particle-hole asymmetry is present in $\Delta\alpha$, the difference between the $\alpha$ values extracted for the two fitting ranges. Figure 3d shows that $\Delta\alpha$ is about 0.08 for the electron side and approaches zero as the hole density increases.

## Discussion
### Influence of the fitting range
As shown above, the MC at sufficiently low magnetic fields can be well-described with the simplified HLN equation (i.e., Eq. (1)), if the MC mainly comes from the quantum interference and MT corrections. At higher magnetic fields the MC is found to deviate slightly from the HLN formula, when the fitting parameters are chosen to provide the best fit for the lowest fields (Fig. 3a). We have also carried out the HLN fits for wider ranges of magnetic field, and the obtained $\alpha$ value decreases with increasing fitting range. This further supports the idea that the EEI corrections are dominated by the terms with different length scales (i.e., in different ranges of $B$). Using a narrower fitting range can reduce the influences of the DOS term, leading to better agreement between the experimental MCs and the HLN equation. However, making the fitting range too narrow would also make the fit unreliable because the MC near the weak field limit approaches a quadratic form, which is unsuitable for extracting two fitting parameters (i.e., $\alpha$ and $B_\varphi$). In the following analyses, we mainly rely on the results obtained by the fitting range of $0-15B_\varphi$, which is narrow enough to avoid significant deviation of the MC curve from the HLN equation, and wide enough to carry out reliable HLN fits. Further discussion of the influences of the fitting range can be found in Supplementary Note 5.

### Values of $\alpha$ at large and small conductances
The observation of $\alpha \approx 1$ (Fig. 2c) at large conductance values ($g \gg 1$) is not surprising for 3D TIs with insulating bulk. If the dephasing lengths for the top and bottom surfaces are close to each other, the surfaces can be considered as two equivalent and independent channels, resulting in $\alpha \simeq (1/2) \times 2 = 1$. This has been demonstrated previously in $Bi_2Se_3$, $(Bi,Sb)_2Te_3$, and BSTS thin films or flakes by many groups[15,16,18–20]. In these studies, $g$ was typically in the range of $7 - 10$, nearly satisfying the requirement for weakly disordered diffusive transport. As shown in Fig. 2a, this is not the case for $V_{BG} < -10$ V, at which $g$ is in a range of $3 - 5$. Importantly, deviation of the HLN

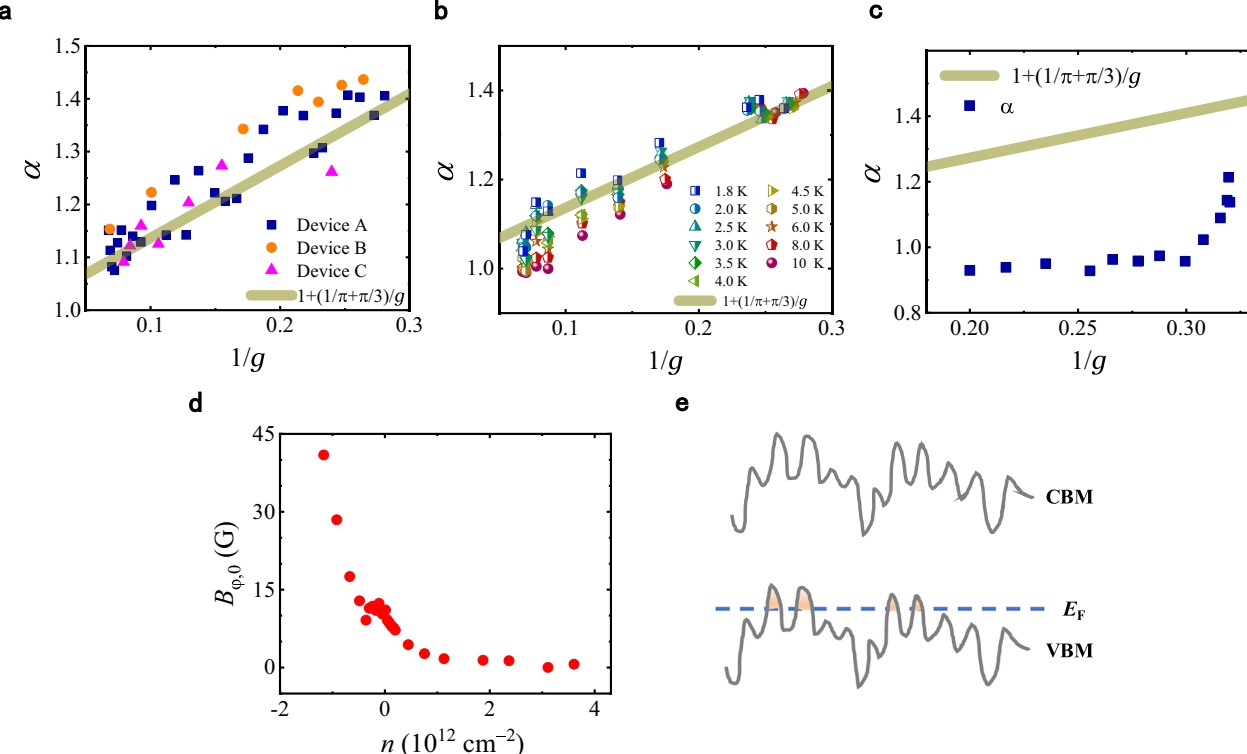

**Fig. 4 | Dependence of prefactor α on 1/g and the role of hole puddles in the bulk. a** α vs. 1/g for three samples (Devices A, B, and C) at T = 1.6 K. **b** α vs. 1/g at various temperatures (1.8–10 K). The data in panels **a** and **b** are only for electron side. **c** α vs. 1/g for the hole side. The solid lines in panels **a**–**c** represent theoretical values given by Eq. (3). **d** Zero-temperature dephasing field plotted as a function of total carrier density. **e** Schematic diagram of the TI bulk bands with strong fluctuations originating from the compensation doping. The filled areas represent nanoscale hole puddles.

prefactor from α = 1 is most pronounced in this region, except for the data points corresponding to substantial hole densities, as depicted in Fig. 2c. The stronger enhancement of α at smaller g-values suggests the importance of the quantum corrections of higher orders in 1/g.

**Nearly linear dependence of α on 1/g**
Let us first focus on the enhancement of α on the electron side, since the conductivity g can be varied across a much larger range than the hole side. Figure 4a depicts α extracted from the low-field HLN fits as a function of 1/g for three devices (Samples A, B, and C) measured at T = 1.6 K. All of them exhibit an approximately linear dependence on 1/g. Moreover, these α values are quite close to the theoretical limit that only includes the quantum interference effects (the first- and second-order corrections) and MT correction with the renormalized coupling, Eq. (1), i.e.,

$$\alpha \approx \alpha_{QI+MT} \simeq 1 + \frac{1}{\pi g} + \frac{\pi}{3g} = 1 + \left(\frac{1}{\pi} + \frac{\pi}{3}\right)\frac{1}{g} \quad (3)$$

The observed agreement of the low-field MC prefactor with the result obtained without the cross-term $\Delta\sigma_{CWAL}(B)$ suggests that, in the range of weak B, this term does not contribute significantly to the B-dependence of the conductivity. This allows us to assume that the B-dependence of $\Delta\sigma_{CWAL}(B)$ is suppressed by the screening by the metallic top-gate in the weak fields satisfying $l_B > d$, whereas the value of the interaction constant $\gamma_c$ in the particle-particle channel is already close to its universal limit (see Supplementary Notes 1 and 5 for details). Figure 4b shows that the α values for a wide range of temperatures (1.8–10 K) are not far away from this theoretical line. The weak T-dependence of α also indicates that the interaction constant in the Cooper channel is already renormalized to its infrared value, since

otherwise the MT correction would contain a $1/\ln^2 T$ prefactor that would make α smaller at lower temperatures[7].

Despite the surprisingly good agreement between the experimental α values and the theoretical ones under the simple assumptions mentioned above, we would like to note that the other contributions (i.e., the DOS correction in the particle-particle channel and the cross-term correction in the particle-hole channel) to the MC cannot be fully excluded for the fitting range of B = 0−15$B_\varphi$, especially for small g values, since the maximum B-fields are comparable to the values of $2\pi B_T$ for the TI samples studied this work (see Supplementary Note 5). Nevertheless, the degree of the overlap between the fitting range and B > $2\pi B_T$ (roughly corresponding to $l_B < l_T$) is a function of lng/g, which decreases monotonically with increasing g. Moreover, the cross-term correction ($\Delta\sigma_{CWAL}$) also has a prefactor proportional to 1/g. This, together with the 1/g-dependence of the second-order interference and the MT corrections, can account for the nearly linear dependence of α on 1/g depicted in Fig. 4.

**Particle-hole asymmetry**
In contrast with the electron side, the values of α are much more reduced for the holes with the same density (Fig. 2c) or conductivity (Fig. 4c). As the second-order interference correction is only dependent on the value of g, the difference between the electron and hole sides must originate from the change in the EEI-related MC corrections. As shown in Fig. 2a, the particle-hole asymmetry in g is quite modest. This indicates that the surface states themselves are unlikely to be solely responsible for the strong asymmetry in α. According the ARPES measurement[29], the Dirac point in BSTS is very close to valence band. It is also known that, there are strong electrostatic potential fluctuations in BSTS due to the de facto compensation doping in this compound. The coexistence of donor- and acceptor-types of defects

is valuable in reducing the bulk conductivity in BSTS, but also results in the formation of nanoscale charge puddles in the bulk[31]. In the high-quality BSTS samples of thin slab geometry, the formation of charge puddles could be suppressed by the screening of the surface states with sufficiently high electron density. However, this is not case for the ambipolar region and nearby ($V_{BG} < -13$ V). Because of the limited screening capability of the surface states and the proximity of the Dirac point to the bulk valence band, the formation of hole puddles in the bulk is unavoidable (Fig. 4e). Nevertheless, the magnitude of chemical potential fluctuations is still considerably smaller than the bulk band gap in the BSTS samples, so the electron puddles in the bulk are very unlikely to coexist with the hole puddles when the Fermi level is near or below the CNP. As the chemical potential is further lowered from the CNP, the number and sizes of the hole puddles are expected to increase, leading to stronger screening effect of the hole puddles and consequently greater suppression of the EEI-related corrections to MC, including the MT correction.

### Low-temperature dephasing

Having established the validity of the HLN equation in describing the low-field MCs, we are now able to discuss the dephasing in the limit of lowest temperatures. The linear $T$-dependence of the dephasing field (rate) is valid down to at least 0.1 K, and extrapolation of the data to $T = 0$ yields a non-zero intercept (Fig. 2d), indicating existence of additional mechanism of dephasing at low temperatures. As depicted in Fig. 4d, the intercept, denoted as $B_{\varphi,0}$, is nearly zero for high electron densities (corresponding to Fermi Energy $E_F > 32$ meV). As the chemical potential is lowered into the ambipolar region and below, $B_{\varphi,0}$ increases nearly monotonically. Since the variation in the conductivity is quite modest in this region (see Fig. 2a), the rapid increase in $B_{\varphi,0}$ with decreasing gate voltage can be mainly attributed to the increase in the $T$-independent contribution to the dephasing rate, which can be evaluated with $1/\tau_{\varphi,0} = 4DeB_{\varphi,0}/\hbar$. This adds on top of the intrinsic $T$-dependent dephasing caused by the EEI[28], resulting in the saturation of linearly extrapolated dephasing rate at $T = 0$. Interestingly, the deviation from the conventional EEI result is much stronger on the hole side of the Dirac point than the electron side. Such an increase in the dephasing rate can be attributed to the hole puddles, which can couple to the surface states via tunneling processes or Coulomb interaction. Interactions with effective two-level systems formed by the states in the vicinity of the surfaces could lead to an enhancement of dephasing at low temperatures[32,33]. This mechanism can cause the dephasing rate to seemingly saturate in a limited temperature range before vanishing at $T \to 0$. It is also noteworthy that similar behavior of the extracted dephasing rate has also been observed in an InAs quantum well tunneling-coupled with a partially populated delta-doping layer[34]. Figure 4d shows that near the CNP, $B_{\varphi,0}$ is more than 10 G, corresponding to a dephasing rate on the order of $10^{11}$ s$^{-1}$. This could be detrimental to the pursuit of delicate quantum transport phenomena near the Dirac point[21–23] (e.g., Majorana zero mode). To this end, searching for higher-quality TI materials and studying the dephasing rate at ultralow temperatures are indispensable for further advancing the field of topological insulators.

In summary, we have shown that the EEI and higher-order quantum interference effects can substantially modify the MC in the 2D systems in the symplectic symmetry class. In BSTS, arguably the most attractive 3D TI material to date, the low-field MC appears to be close to the sum of the interference and the MT corrections, and hence can be described by the HLN equation rather well. The dephasing field extracted from the fit with the HLN formula can provide meaningful information on electron dephasing, and our data suggests that the dephasing rate is enhanced substantially by additional to EEI mechanisms at temperatures below a few K. The MC in higher magnetic field is, however, more involved with the contributions of other

EEI effects, showing a hierarchy of multiple length scales. Our work also demonstrates that the detailed study of the MC enables us to obtain a deep knowledge of the electronic states in BSTS, for instance, the mechanism underlying the particle-hole asymmetry and the saturation tendency of dephasing time in the low-temperature limit. This should be beneficial to the study of other types of TI or more generally, low dimensional systems of the symplectic symmetry class.

## Methods

### Material growth

(Bi,Sb)$_2$(Te$_x$,Se$_{1-x}$)$_3$ (BSTS) single crystals used in this work were synthesized with a flux method. Bi, Sb, Te, and Se (all with a purity level of 99.999%) were used as starting materials. They were mixed at a molar ratio of 1:1:1:2 in a glove box filled with high-purity argon. The mixture was then sealed in a quartz tube, heated to 850 °C, and kept at this temperature for an hour, before slowly cooling down to the room temperature over a time span of 2 weeks.

### Device fabrication

For the transport device fabrication, pre-patterned electrodes were first prepared on SiO$_2$/Si substrates with standard photolithography, followed by thermal or electron beam deposition of a Pd/Au (2 nm/15 nm thick) bilayer. The Scotch tape method was used to exfoliate the BSTS single crystals into 20–60 nm thick microflakes onto a substrate. The BSTS microflakes were then picked up by a polypropylene carbonate (PPC) stamp, and released onto a substrate with the pre-patterned electrodes by using the dry transfer method. The above exfoliation/pickup/release procedures were repeated to obtain a large $h$-BN microflake for encapsulating the BSTS microflake. The $h$-BN layer also serves as the top-gate dielectric, which can offer a carrier-density tuning-range comparable to the bottom-gate dielectric (i.e., the 300 nm thick SiO$_2$ layer, with the heavily-doped Si substrate working as the bottom-gate electrode). Subsequently, a Pd (2 nm)/Au (80 nm) bilayer (top gate) was evaporated onto the $h$-BN microflake, on which the top-gate electrode regions were defined by e-beam lithography. Shown in Fig. S2 are optical images of samples A–C.

### Electron transport measurements

Before the electron transport measurements, all electrical contacts were checked at liquid helium temperatures. The $I$–$V$ characteristics are linear, and the contact resistances are usually a few kΩ. The transport measurements were performed in a $^3$He cryostat, as well as a top-loading dilution refrigerator, with the standard low-frequency lock-in technique. The excitation current was usually set at 10–100 nA to avoid Joule heating or other spurious effects. The top and bottom-gate-voltages were supplied with a pair of source-measurement units with sub-pA leakage current detecting capability.

## Data availability

The data supporting the findings of this study are available provided in the figures and Supplementary Information, and are publicly available at https://doi.org/10.5281/zenodo.7733705. Additional data are available from the corresponding authors upon request.

## Code availability

The computer codes used for the data analysis and numerical simulation can be obtained by making request to the corresponding authors.

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

## Acknowledgements

We are grateful to Zhichuan Wang for assistance in numerical simulations, and I. Burmistrov, G. Minkov, P. Ostrovsky, H. Y. Xie, and Y. Xu for valuable discussions. Y.L. acknowledges financial support by the National Natural Science Foundation of China (Grant No. 11961141011), the Strategic Priority Research Program of Chinese Academy of Sciences (Grant No. XDB28000000), and the National Key Research and Development Program of China (Grants No. 2016YFA0300600 and No. 2022YFA1403403). D.G. acknowledges funding support by ISF-China 3119/19 and ISF 1355/20. Z.L. acknowledges support from National Natural Science Foundation of China (Grant No. 12204520) and the Youth Innovation Promotion Association of the Chinese Academy of Sciences (Grant No. 2021008).

## Author contributions

Y.L. and D.G. initiated the project. Z.L. carried out the growth of BSTS single crystals. G.S. and R.C.Z. fabricated the devices. G.S. and F.G. performed the electron transport measurements. G.S., I.G., D.G., and Y.L. analyzed the data and prepared the manuscript with contributions from all authors.

## Competing interests

The authors declare no competing interests.
