## [Peer Review File · Nature Communications]

Quantum corrections to the magnetoconductivity of surface states in three-dimensional topological insulatorsREVIEWER COMMENTS

Reviewer #1 (Remarks to the Author):

The article is about study of transport mechanism by analysing magneto transport properties of BSTS topological insulators. The effect of EEI and higher order quantum interference effect in particle-particle and particle-hole channels modifies the overall MC. The article is well written and explains all the facts that are found in their transport properties but before accepting the article the author should reconsider few things in the current MS which are highlighted below.

The magneto-conductivity in low magnetic field regime is explained HLN eq. in which MC is function of B_ϕ/B , but in eq.1, γ is the function of B/B_ϕ , explain.

In eq. S7, the logarithmic correction to MC is defined in terms of singlet channel and triplet channel.

How to identify and separate effects of these channels in given formula.

In eq. S8, both the third and fourth logarithmic terms are same, then why only the fourth logarithmic term is suppressed in high magnetic field limit and eq. S8 converges to eq. S7. Explain

From fig. 4c, the values of α are found to be increased in case of holes as compared to that of electrons, but it is mentioned in text (line 243) that α value reduced.

The HLN does not fully explain the behaviour of magnetoresistance in entire magnetic field range. So, in addition to the explained effect in current MS, the author should reconsider the same by modifying the theoretical modelling through quadratic and linear terms as mentioned in recent published articles such as

"Comprehensive Analysis for the High Field Magneto-conductivity of Bi₂Te₃ Single Crystal", Physica B, 609, 412759 (2021) and "Origin of large magnetocapacitance in K_{0.5}Na_{0.5}NbO₃ / La_{0.67}Sr_{0.33}MnO₃ superlattices", Phys. Rev. B 106, 155403 (2022).

Reviewer #2 (Remarks to the Author):

G. Shi et al. report the quantum corrections to the magnetoconductivity (MC) of topological surface states in BSTS flakes. The authors have analyzed the MC of BSTS flake with tunable Fermi level position in different magnetic field region qualitatively. The different contributions of the MC in BSTS have been recognized. The low-field MC is proposed to be dominated the quantum interference and the Maki-Thompson (MT) corrections and high-field MC is contributed from the extra complex EEI corrections. The physical insight into the underlying mechanisms of the MC in topological insulators is of tremendous significance in condensed matter physics. I believe the attempt in the manuscript is interesting and is worthy more scientific concerns. However, the conclusion in the manuscript should be further strengthened before considering its publications in Nature Communications.

1.The HLN fitting is usually applicable in the regions $B_{so} > B_\phi$, where B_{so} and B_ϕ are the spin-orbit characteristic field and characteristic dephasing field. Have the authors considered such preliminary conditions for the fitting region selection when conducting the HLN fitting?

2.The authors claimed the extracted α is linearly dependence on the $1/g$ in Fig. 4a and 4b. However, such statement is not convincing enough from the presented figures for all three devices in Fig. 4a and temperature above 3.0 K in Fig. 4b. This strongly diminishes the novelty the central results of the manuscript.

3.It is strange to me that the distinct behaviors of hole side (Gate A) and electron side (Gate C) in the vicinity of the Dirac point are related to the formation of hole puddles. Although the Fermi level closes to the valence band, the bandgap in BSTS is quite small, the electron puddles should be also existed.

4.Some typos should be revised thoroughly, and more experimental results should be presented, at least in the supplementary materials, such as the measured MC of BSTS flakes with different

thicknesses in both perpendicular and parallel fields.

Reviewer #3 (Remarks to the Author):

In this manuscript, the authors extend the theory of quantum conductance correction to two-dimensional electron system with the symplectic symmetry, study the corresponding physics in dual-gated topological insulator devices. It is found that the weak field magnetoconductance corrections can be enhanced significantly by the second-order quantum interference and the electron-electron interaction effects, in contrast to suppression for systems with orthogonal symmetry. The results look interesting.

There are some concerns:

The topic has been studied extensively since the discovery of topological insulator. It is a question whether we can simply apply the HLN formula to fit all experimental data. In the hole side, the values of α are reduced to be smaller than one. According to Eq. (3), α should always be greater than one. Even if the EEI-related MC corrections are suppressed by the formation of hole puddles, the contribution from the second-order correction is not negative. In the electron sides, the values of α are greater than one. Can the authors give more explanation on this discrepancy?

There are several different effective fields, i.e., B_ϕ , B_T , and B_{tr} . The EEI-related quantum correction depends on B_T and B . Eq. (2) requires $B \ll B_T$. However, according to Table IV in the Supplementary Materials B_T is much smaller than B_{max} . Eq. (2) does not hold anymore, and DOS correction and "cross-term" MC correction will also contribute to the conductivity correction. Hence, there are some inconsistencies between the theory and fitting parameters.

In the hole side, the Fermi energy may intersect with the bulk sub-bands, Does the bulk sub-bands contribute to the MC correction?

Response to the reports of referees

We thank all three referees for carefully reviewing our manuscript and making constructive comments, which are valuable in improving the quality of our manuscript. Please see below for the response to each of the technical comments.

Reviewer1:

The article is about study of transport mechanism by analysing magneto transport properties of BSTS topological insulators. The effect of EEI and higher order quantum interference effect in particle-particle and particle-hole channels modifies the overall MC. The article is well written and explains all the facts that are found in their transport properties but before accepting the article the author should reconsider few things in the current MS which are highlighted below.

1. The magneto-conductivity in low magnetic field regime is explained HLN eq. in which MC is function of B_\perp/B , but in eq.1, Y is the function of B/B_\perp , explain.

Reply: The definition of the Y -function originates from Ref. 7 (Minkov et al., 2004), namely $Y(x) = \Psi\left(\frac{1}{2} + \frac{1}{x}\right) + \ln(x)$, where $x = \frac{B}{B_\perp}$. In fact, it is easy to check that this is fully consistent with the original HLN paper, in which Eq. 6 reads

$$\Delta\sigma = -\frac{\alpha e^2}{2\pi^2\hbar} \left[\ln\left(\frac{1}{\tau_\varepsilon a}\right) - \Psi\left(\frac{1}{2} + \frac{1}{\tau_\varepsilon a}\right) \right].$$

Identifying the scale τ_ε of the HLN with the dephasing time τ_ϕ and using the definition of the quantity $a = \frac{4DeB}{\hbar c}$, one finds $\tau_\phi a = \frac{B}{B_\perp}$. Thus, the MC function in the HLN paper can be equivalently expressed through $x = \frac{B}{B_\perp}$. Note that the parameter α is defined with a sign opposite to the original HLN paper (i.e., Ref. 6 of the current manuscript) for the convenience in discussing the weak antilocalization effect.

2. In eq. S7, the logarithmic correction to MC is defined in terms of singlet channel and triplet channel. How to identify and separate effects of these channels in given formula.

Reply: The second-loop interference correction (S7) is represented as

$$g(L) = g^0 - \frac{1+3}{2\pi^2 g^0} \ln\left(\frac{L}{l_{tr}}\right) = g^0 - \frac{1}{2\pi^2 g^0} \ln\left(\frac{L}{l_{tr}}\right) - \frac{3}{2\pi^2 g^0} \ln\left(\frac{L}{l_{tr}}\right)$$

In other words, the term coming with a coefficient 1 corresponds to a singlet, and a term coming with a coefficient 3 to a triplet.

These terms can be separated by considering systems with a finite spin-orbit interaction length L_{so} . When $L > L_{so}$, one replaces L with L_{so} in the triplet term, while the singlet term is unaffected by the spin-orbit interaction. This will result in a change of the temperature dependence of conductivity in the unitary ensemble (strong magnetic field) stemming from this correction. At high temperatures, when $L = L_{\varphi} < L_{so}$, all four channels will contribute to the logarithmic temperature dependence of conductivity, while at lowest temperatures, $L_{\varphi} < L_{so}$, only the singlet channel will play a role.

It is worth noting that in Supplemental Note 1, section A, equations S2-S13 (including Eq. S7) are given for topologically trivial metals, such as the InGaAs quantum well studied in Ref. 7, in which both the singlet and triplet terms contribute to the logarithmic conductivity corrections. However, for the TI surface states, only the singlet term remains, because of the spin-helical electronic structure described by the massless Dirac equation. In this sense, the TI surface states may be regarded as a system in the strong spin-orbit-coupling limit, with $L_{so} \rightarrow 0$, so that there are no logarithmic triplet terms. This situation is described in Sections B-D of Supplementary Note 1, where the formulae for various corrections to the conductance of TI surface states (no triplet channel) are presented.

3. In eq. S8, both the third and fourth logarithmic terms are same, then why only the

fourth logarithmic term is suppressed in high magnetic field limit and eq. S8 converges to eq. S7. Explain

Reply:

$$g(L) = g^0 + \frac{1-3}{\pi} \ln\left(\frac{L}{l_{tr}}\right) - \frac{1+3}{2\pi^2 g^0} \ln\left(\frac{L}{l_{tr}}\right) + \frac{1+3}{2\pi^2 g^0} \ln\left(\frac{L}{l_{tr}}\right)$$

\updownarrow

Cooperon channel
(one loop)

\updownarrow

Diffuson channel
(two loops)

\updownarrow

Cooperon channel
(two loops)

In this expression, L denotes the RG running infrared scale. Note that, in the presence of the magnetic field and the spin-orbit interaction, the problem has more than one infrared cut-off. In particular, the localization corrections involving the Cooperon channel are sensitive to the magnetic field (the second and the fourth terms in the equation above). Therefore, the scale controlled by the magnetic field, l_B , plays a role of an infrared cutoff in these terms. This is in contrast with the term coming from two-loop diffusion correction (the third term in the equation above), which does not involve Cooperons. These terms are unaffected by the magnetic field in this range of parameters. Therefore, the scale l_B can never appear as RG scale in the diffuson term. As a result, the cancellation between the two-loop Diffuson and Cooperon contributions (the third and fourth terms) only happens at zero magnetic field. At any finite magnetic field these terms no longer cancel each other, giving rise to the finite magnetoresistance.

4. From fig. 4c, the values of α are found to be increased in case of holes as compared to that of electrons, but it is mentioned in text (line 243) that α value reduced.

Reply: In Fig. 4c, the experimental α values for holes (blue squares) are in fact lower than those for electrons, which roughly follow the theoretical curve (Eq. (3), yellow-grey line in the figure, see also Fig. 4a and Fig. 2c). The text (line 243) is hence consistent with the figure.

The HLN does not fully explain the behaviour of magnetoresistance in the entire magnetic field range. So, in addition to the explained effect in current MS, the author should reconsider the same by modifying the theoretical modelling through quadratic and linear terms as mentioned in recent published articles such as “Comprehensive Analysis for the High Field Magneto-conductivity of Bi₂Te₃ Single Crystal”, *Physica B*, 609, 412759 (2021) and “Origin of large magnetocapacitance in K_{0.5}Na_{0.5}NbO₃ / La_{0.67}Sr_{0.33}MnO₃ superlattices”, *Phys. Rev. B* 106, 155403 (2022).

Reply: We thank the referee for attracting our attention to these two papers on magnetoresistance (MR). The quadratic and linear MRs have also been observed in topological insulators by our group and by many others. Among the topological insulators, the most pronounced linear MR was observed, in a very wide range of magnetic fields, in Bi₂Te₃. However, it was also shown that the linear MR crosses over to a parabolic dependence at sufficiently low fields (typically of the order 0.1 T), when the weak antilocalization effect is suppressed by raising the temperature to about 10 K (see, for instance, Qu et al., *Science* 329, 821 (2010)).

TABLE R1 Comparison of fitting parameters obtained by the HLN fits of the magnetoconductance (MC) data before (α , B_ϕ) and after (α' , B'_ϕ) the quadratic magnetoresistance background is removed. The corresponding MC data is shown in Fig. S6 in the supplementary materials.

Gate-voltage point	α	B_ϕ (G)	α'	B'_ϕ (G)
A	1.023	34.24	1.022	34.21
B	1.286	27.32	1.285	27.30
C	1.403	23.31	1.403	23.32
D	1.264	14.13	1.262	14.11
E	1.082	8.148	1.082	8.162

For BSTS, however, no clear evidence has been reported so far for this type of linear MR, despite extensive studies. The quadratic MR is thus expected to exist in a wider range of magnetic fields in BSTS than Bi₂Te₃. In this work, we are mainly concerned with the MC corrections related to the weak antilocalization effect. The

related data analyses are limited to $B < 0.25$ T. The low-field quadratic MR background are obtained by fitting the MR data taken at high magnetic fields (4-9 T). After this background is subtracted, the low-field MR data is converted to the MCs and fitted to the HLN equation (or its generalization). Table R1 shows that the fitting results are barely modified by the removal of quadratic background. We added a new figure in the supplementary materials (Supplementary Fig. S6) to address this issue. The paper on Bi_2Te_3 (Physica B **609**, 412759 (2021)) is also added as a reference.

Reviewer2:

G. Shi et al. report the quantum corrections to the magnetoconductivity (MC) of topological surface states in BSTS flakes. The authors have analyzed the MC of BSTS flake with tunable Fermi level position in different magnetic field region qualitatively. The different contributions of the MC in BSTS have been recognized. The low-field MC is proposed to be dominated the quantum interference and the Maki-Thompson (MT) corrections and high-field MC is contributed from the extra complex EEI corrections. The physical insight into the underlying mechanisms of the MC in topological insulators is of tremendous significance in condensed matter physics. I believe the attempt in the manuscript is interesting and is worthy more scientific concerns. However, the conclusion in the manuscript should be further strengthened before considering its publications in Nature Communications.

1. The HLN fitting is usually applicable in the regions $B_{so} > B_\phi$, where B_{so} and B_ϕ are the spin-orbit characteristic field and characteristic dephasing field. Have the authors considered such preliminary conditions for the fitting region selection when conducting the HLN fitting?

Reply: For 3D TI surface states, the spin-momentum locking generates a Berry phase π . This corresponds to a conventional semiconductor system with infinite spin-orbit coupling strength. Therefore, the prerequisite $B_{so} > B_\phi$ is satisfied.

2. The authors claimed the extracted α is linearly dependence on the $1/g$ in Fig. 4a and 4b. However, such statement is not convincing enough from the presented figures for all three devices in Fig. 4a and temperature above 3.0 K in Fig. 4b. This strongly diminishes the novelty the central results of the manuscript.

Reply: We thank the referee for the careful examination of our data. The data shown Fig. 4a supports an approximate linear dependence on $1/g$, even though the slope is different from the Eq. (3) (grey line) to some extent. As to Fig. 4b, the deviation from the linear dependence gets a bit more obvious at higher temperatures and larger conductances (smaller $1/g$ values). Nevertheless, the overall characteristic of the low- T data from all samples seems to be consistent with the linear dependence on $1/g$, at least approximately. From the theoretical point of view, Eq. (3) is a somewhat oversimplified representation of the higher-order quantum corrections to the MC. The contributions of the interaction corrections, which include the Maki-Thompson (MT) and DOS terms in the particle-particle channels, as well as the cross-term correction in the particle-hole channel, are condensed into a single term with a prefactor approximately equal to $\pi/3$ (equivalent to the MT correction at the infrared limit). Because of the competition between several different length scales, the deviation of α from this value is not surprising, but very challenging to describe quantitatively. At large g -values, where the deviation is most obvious (but still less than 0.1), we speculate the screening effect by the surface states themselves might be relevant, since the electron densities are on the order of $1E12\text{ cm}^{-2}$. Another possibility is the appearance of some electron puddles at such high Fermi levels (but still in the bulk band gap) due to disorder and thermal activation. As a result, the electron-electron interaction deviates more from the long-range Coulomb interaction (the infrared limit). It is also noteworthy that the magnitude of deviation from the linear dependence is comparable to the size of error bars of the extracted α values, so this issue could be better addressed in future studies of higher quality TI samples.

In this manuscript, all the quantum corrections beyond the standard weak antilocalization (HLN) theory, are related to the corrections of next order in $1/g$ in the

scaling theory. Our extensive study shows that, despite coexistence of multiple interaction effects with different length scales, the HLN fits remain largely valid for the case of small g -values (i.e., beyond the weak disorder regime), at least at low magnetic fields. Given that the α value is independent of $1/g$ in the standard HLN theory, a linear dependence of α on $1/g$ is hence reasonable for the corrections of the next order in $1/g$. We have modified the main text as well as Supplementary Note 5 to explain the $1/g$ dependence more clearly.

3. It is strange to me that the distinct behaviors of hole side (Gate A) and electron side (Gate C) in the vicinity of the Dirac point are related to the formation of hole puddles. Although the Fermi level closes to the valence band, the bandgap in BSTS is quite small, the electron puddles should also exist.

Reply: We thank the referee for this comment. Based on the positions of Hall resistance extrema, the fluctuations in the chemical potential are estimated to be about 0.05 eV, much smaller than the bulk band gap of BSTS (0.25-0.30 eV). Since the Dirac point is very close to the bulk valence band, formation of electron puddles is unlikely if the Fermi level is drawn close to the bulk valence band. We have added a sentence in the main text to exclude the possibility of electron puddles in this situation. It reads *“Nevertheless, the magnitude of chemical potential fluctuations is still considerably smaller than the bulk band gap in the BSTS samples, so the electron puddles in the bulk are very unlikely to coexist with the hole puddles when the Fermi level is near or below the CNP.”*

4. Some typos should be revised thoroughly, and more experimental results should be presented, at least in the supplementary materials, such as the measured MC of BSTS flakes with different thicknesses in both perpendicular and parallel fields.

Reply: We thank the referee for pointing this out. We have carefully checked the manuscript, including the Supplemental Information, and corrected all the typos we could find. We have also added the MC data taken in both perpendicular and parallel

magnetic fields from several samples (see Supplementary Figs. S9-S11).

Reviewer 3:

In this manuscript, the authors extend the theory of quantum conductance correction to two-dimensional electron system with the symplectic symmetry, study the corresponding physics in dual-gated topological insulator devices. It is found that the weak field magnetoconductance corrections can be enhanced significantly by the second-order quantum interference and the electron-electron interaction effects, in contrast to suppression for systems with orthogonal symmetry. The results look interesting.

There are some concerns:

1. The topic has been studied extensively since the discovery of topological insulator. It is a question whether we can simply apply the HLN formula to fit all experimental data. In the hole side, the values of α are reduced to be smaller than one. According to Eq. (3), α should always be greater than one. Even if the EEI-related MC corrections are suppressed by the formation of hole puddles, the contribution from the second-order correction is not negative. In the electron sides, the values of α are greater than one. Can the authors give more explanation on this discrepancy?

Reply: Indeed, the α values smaller than one on the hole side cannot be attributed to the EEI-related corrections. Since this effect is absent on the electron side, the populated states in the bulk valence band should be responsible. According to Garate and Glazman (PRB 86, 035422 (2012)), the bulk states, if modeled as a massive 2D Dirac fermion system, could produce weak localization effect and hence positive MC. Therefore, the coupling of the Dirac surface states to the bulk states can in principle account for the reduction in the α value. However, a quantitative treatment of this effect is beyond the scope of this work due to the presence of the long-range disorder, which makes the electron system strongly inhomogeneous. We have added the Garate-Glazman paper as Ref. 14 in the revised manuscript.

2. There are several different effective fields, i.e., B_ϕ , B_T , and B_{tr} . The EEI-related quantum correction depends on B_T and B . Eq. (2) requires $B \ll B_T$. However, according to Table IV in the Supplementary Materials B_T is much smaller than B_{max} . Eq. (2) does not hold anymore, and DOS correction and “cross-term” MC correction will also contribute to the conductivity correction. Hence, there are some inconsistencies between the theory and fitting parameters.

Reply: We thank the referee for making this keen point. In the main text, the statement we made on Eq. (2) was a bit oversimplified. A more precise expression for the MT correction is $\Delta\sigma_{MT} \simeq -\frac{\pi^2}{6}\gamma_c \left[Y\left(\frac{B}{B_\phi}\right) - Y\left(\frac{B}{2\pi B_T}\right) \right]$ (Eq. A6 in Ref. 7), where $Y(x) = \Psi\left(\frac{1}{2} + \frac{1}{x}\right) + \ln(x)$. The condition for Eq. (2) to be accurate is hence $B \ll 2\pi B_T$. In fact, as shown in a new figure in the Supplementary Information (Fig. S7, duplicated below), the difference between these two equations is very small for the field range $B = 0-15B_\phi$.

Fig. R1 Comparison between two forms of the MC correction by the Maki-Thompson (MT) term in the Cooperon channel. The black curve plots the function $F(x) = -Y(x)$, and the red one represents $F(x) = -\left[Y(x) - Y\left(\frac{x}{15.7}\right) \right]$. The former is valid for the low magnetic fields, while the latter holds for a wider range of magnetic fields, and hence closer to the experimental observation. At gate-voltage point C (see Fig. 2a), $B_T \approx 2.5B_\phi$, which gives a denominator in the second term of $2\pi B_T/B_\phi \approx 15.7$.

With the help from the DOS term in the particle-particle channel, and suppression of the cross-term correction in the particle-hole channel by the metallic top-gate, the

total MC correction due to the EEI effects can be described by Eq. (2) quite well in sufficiently low magnetic fields, as demonstrated by the MC data in Figs. 2 & 4, as well as in the Supplementary materials. We have modified Supplementary Note 5 substantially to address this issue in greater detail. The related part in the main text is also changed for better clarity.

3. In the hole side, the Fermi energy may intersect with the bulk sub-bands, Does the bulk sub-bands contribute to the MC correction?

Reply: The contribution of the bulk states has been discussed in the response to the first comment, i.e., the origin of α values smaller than one. Here, we would like to discuss this issue in further detail. When the Fermi level is very not very low, for instance, near the charge neutral point, the valence band is only populated in some regions of the bulk, leading to the formation of hole puddles. The top and bottom surfaces are still separated by an insulating bulk layer. These puddles do not contribute directly to the conductance, but they help to suppress the Coulomb interaction via the screening effect and cause electron dephasing either by tunneling or electrostatically coupling to the surface states. When the Fermi level is drawn lower, an insulator-to-metal transition will take place first in a thin layer near the surface. When the gate voltage gets more negative, the insulating layer in the middle will become thinner, allowing for electron tunneling or eventually direct conductance between the top and bottom layers (containing the surface states and nearly populated bulk states). In this case, α values substantially smaller than 1 are expected due to the inter-channel coherent coupling. Such physics has been studied extensively (e.g., Refs. 14-20).

REVIEWERS' COMMENTS

Reviewer #1 (Remarks to the Author):

The MS is now ready from my side for publication by Nat. Comm. Authors have answered my queries to a satisfactory level.

Reviewer #2 (Remarks to the Author):

All my concerns have been well addressed, I, therefore, support its publication on Nature Communications.

Response to Reviewers' Comments

Manuscript ID: NCOMMS-22-40087B

REVIEWERS' COMMENTS

Reviewer #1 (Remarks to the Author):

The MS is now ready from my side for publication by Nat. Comm. Authors have answered my queries to a satisfactory level.

Reviewer #2 (Remarks to the Author):

All my concerns have been well addressed, I, therefore, support its publication on Nature Communications.

REPLY: We thank both referees for their time and efforts, which have helped us improve the quality of our manuscript substantially.